# Assessing ascertainment bias in atrial fibrillation across US minority groups

**Lars Hulstaert**[1]\*, **Amelia Boehme**[2], **Kaitlin Hood**[3], **Jennifer Hayden**[3], **Clark Jackson**[2], **Astra Toyip**[2], **Hans Verstraete**[4], **Yu Mao**[3], **Khaled Sarsour**[3]

**1** R&D Data Science & Digital Health, Janssen-Cilag GmbH, Neuss, North Rhine-Westphalia, Germany, **2** Aetion Inc, New York, New York, United States of America, **3** R&D Data Science & Digital Health, Janssen Pharmaceuticals, Titusville, New Jersey, United States of America, **4** R&D Data Science & Digital Health, Janssen Pharmaceutica NV, Beerse, Antwerp, Belgium

\* lhulsta1@its.jnj.com

## Abstract

The aim of this study is to define atrial fibrillation (AF) prevalence and incidence rates across minority groups in the United States (US), to aid in diversity enrollment target setting for randomized controlled trials. In AF, US minority groups have lower clinically detected prevalence compared to the non-Hispanic or Latino White (NHW) population. We assess the impact of ascertainment bias on AF prevalence estimates. We analyzed data from adults in Optum's de-identified Clinformatics® Data Mart Database from 2017–2020 in a cohort study. Presence of AF at baseline was identified from inpatient and/or outpatient encounters claims using validated ICD-10-CM diagnosis algorithms. AF incidence and prevalence rates were determined both in the overall population, as well as in a population with a recent stroke event, where monitoring for AF is assumed. Differences in prevalence across cohorts were assessed to determine if ascertainment bias contributes to the variation in AF prevalence across US minority groups. The period prevalence was respectively 4.9%, 3.2%, 2.1% and 5.9% in the Black or African American, Asian, Hispanic or Latino, and NHW population. In patients with recent ischemic stroke, the proportion with AF was 32.2%, 24.3%, 25%, and 24.5%, respectively. The prevalence of AF among the stroke population was approximately 7 to 10 times higher than the prevalence among the overall population for the Asian and Hispanic or Latino population, compared to approximately 5 times higher for NHW patients. The relative AF prevalence difference of the Asian and Hispanic or Latino population with the NHW population narrowed from respectively, -46% and -65%, to -22% and -24%. The study findings align with previous observational studies, revealing lower incidence and prevalence rates of AF in US minority groups. Prevalence estimates of the adult population, when routine clinical practice is assumed, exhibit higher prevalence differences compared to settings in which monitoring for AF is assumed, particularly among Asian and Hispanic or Latino subgroups.

**Data Availability Statement:** The source data for this study were licensed by Johnson & Johnson from Optum, and hence we are not allowed to share the licensed data publicly. However, the same data used in this study are available for

purchase by contracting with the database owner, Optum (contact at: https://www.optum.com/business/life-sciences/real-world-data/claims-data.html). The authors did not have any special access privileges that other parties who license the data and contract with Optum would not have.

**Funding:** The authors received no specific funding for this work. LH, KH, JH, HV, YM & KS are full-time salaried employees of Janssen Research & Development, a pharmaceutical company of Johnson & Johnson. AB, CJ & AT are full-time salaried employees of Aetion, Inc. The specific roles of these authors are articulated in the 'author contributions' section. The funders had no role in study design, data collection and analysis, decision to publish, or preparation of the manuscript.

**Competing interests:** LH, KH, JH, HV, YM & KS are employees of Janssen Research and Development, a unit of Johnson and Johnson family of companies. The work on this study was part of their employment. AB, CJ & AT & are paid employees of and shareholders in Aetion, Inc., a company that makes software for the analysis of real-world data. This does not alter our adherence to PLOS One policies on sharing data and material.

# Introduction

## Health disparities in the US

Population subgroups that face health disparities are often underrepresented in clinical research, including clinical trials [1–4]. The bias introduced by underrepresentation of certain populations in randomized controlled trials (RCTs) can result in the inability to apply trial findings to the underrepresented patient subgroups. These understudied populations may experience a different response to the intervention of interest due to variation in disease-specific risk factors, disparities in health care access or pathogenesis differences [5].

## Underrepresentation in clinical trials

In the United States, members of racial and ethnic minority groups have historically experienced inequities in healthcare access and quality; and continue to be underrepresented in clinical trials [6]. Race and ethnicity are proxies for populations that have been historically undertreated or mistreated due to issues with access to care, provider mistrust, socioeconomic status and historic injustices. The importance of representation of minority groups in RCTs is underscored by the passing of the National Institute of Health (NIH) Revitalization Act by Congress in 1993 [7]. This law encourages enrollment of underrepresented populations in RCTs with the goal of increased representation in clinical research and evidence generation. Despite these efforts, ensuring representation and reducing racial and ethnic disparities in clinical research continues to be a challenge across many therapeutic areas [3, 8–12].

Recent guidelines published by the United States (US) Food and Drug Administration (FDA) expand on methods and approaches for increasing representation in RCTs, with the goal of reducing racial and ethnic health disparities [13]. These approaches include establishing subgroup enrollment targets, increasing access to RCT participation, and broadening eligibility criteria. Having more accurate estimates of incidence and prevalence in underrepresented populations can better inform target enrollment goals for these patients. However, setting appropriate enrollment targets for RCTs is challenging when the underlying incidence and prevalence of diseases of interest among racial and ethnic subgroup populations is not well estimated. The FDA suggests leveraging real-world data (RWD) to estimate the incidence and prevalence of health outcomes within racial and ethnic groups, to develop diversity enrollment targets in RCTs [14]. RWD provides an opportunity to assess racial and ethnic differences and disparities in healthcare delivery and outcomes; these epidemiologic findings will allow for real-world representative enrollment targets, compared to simpler overall demographic estimates [15–17].

## Assessing incidence and prevalence estimates across racial and ethnic groups with RWD

When assessing racial and ethnic differences in incidence and prevalence in RWD, the aim is to characterize true differences in disease presentation amongst subgroups and exclude differences that originate from healthcare delivery bias. Patient level healthcare data are not a random sample from the overall population and tend to be biased towards patients that have access to healthcare. One such example is ascertainment bias, a sampling bias that can result from differences in healthcare access, provider bias, and healthcare system behaviors. This is particularly true for, but not limited to, insurance claims based RWD, where communities which historically have been subject to health disparities can be under- or misrepresented [18–20]. Ascertainment bias can be introduced by different people during a healthcare intervention, from the person receiving the intervention to the person administering the intervention.

Groups that are under-ascertained may be less likely to seek or receive care, even when comparably insured, due to other factors such as accessibility, language, cultural and knowledge barriers, systemic racism, and discrimination. Additionally, even when they do seek care, these groups can still be misdiagnosed due to provider bias [21].

## Atrial fibrillation (AF) in racial and ethnic groups in the US

AF prevalence and incidence is growing due to the aging population, increased risk factors, and improved detection [22]. AF is often diagnosed when patients present with symptoms or incidentally during routine medical examinations, through systematic screening, or following a medical event (e.g., ischemic stroke). Early detection of AF is critical to reduce the risk of complications; however, AF is often paroxysmal (intermittent), and diagnosis remains challenging. Numerous observational studies have studied risk, incidence, and prevalence estimates of AF across racial and ethnic groups [23–39]. Although Black or African American populations show higher rates of traditional cardiovascular risk factors the incidence and prevalence rates of AF remain lower when compared to NHW populations. This contradiction is often referred to as the 'AF paradox' [40]. While the low prevalence of AF in the Black African American population has been thoroughly studied, with several proposed hypotheses for this phenomenon ranging from ascertainment bias to genetic and environmental risk factors, limited research has been conducted to examine the lower AF incidence and prevalence rates within the Asian and Hispanic or Latino population.

Generally, studies can be divided into two groups based on AF detection method. The clinical-based detection group refers to studies where AF cases were identified with routine clinical methods. This includes validating the AF diagnosis by ECGs obtained during protocol-driven events or at study baseline, through hospitalization records, EHR, administrative claims, discharge codes, death certificates, or self-reporting by the patient. The monitoring-based detection group refers to studies where AF cases were identified through ambulatory monitoring, e.g., 48-hour or 14-day ambulatory ECG recordings or when ECG monitoring is presumed through clinical events, e.g., patients with a pacemaker or patients who experienced a stroke event. A detailed overview of studies in terms of the enrollment period, enrolled population, and AF detection method is available in S1 Table.

This study aims to estimate the incidence and prevalence within racial and ethnic groups for atrial fibrillation using a large-scale US-based claims RWD source, to support evidence-based diversity enrollment target setting. The incidence and prevalence measures are explored both in the overall adult population, as well as a targeted population that has been systematically tested for AF after an ischemic stroke event. The hypothesis of this study is that populations undergoing systematic AF screening for medical cause will provide incidence/prevalence estimates closer to the "true" disease occurrences. Additionally, we aim to better understand the impact of ascertainment bias on prevalence rates by computing prevalence ratios and relative differences across racial and ethnic groups, both in a clinical-based and monitoring-based detection setting in claims data.

## Materials and methods

### Study design and setting

This non-interventional cohort study was designed to evaluate the annual incidence and prevalence of AF in an overall patient population as well as a subset with a history of ischemic stroke. Annual incidence was evaluated for each calendar year from 2017–2020. Prevalence was evaluated for the period 2017–2020. The study used Optum's de-identified Clinformatics® Data Mart Database (CDM) from 1 January 2016 to 31 December 2020. This US-

based longitudinal dataset consists of administrative health claims for approximately 70 million de-identified patients sourced from a large national managed care company in the US, and includes data on member eligibility and demographics, inpatient and outpatient claims, outpatient lab tests, and socioeconomic information. The patient's name and geography are used by the data vendor to map the patient to one of five race and ethnicity categories: Asian, Black or African American, Hispanic or Latino, non-Hispanic or Latino White or Unknown/ Other, aligned with the U.S. OMB (U.S. Office of Management and Budget standards) classifications. CDM data only contains de-identified health information as described by the HIPPA Privacy Rule. No direct identifiers of individuals or providers are included. The use of the CDM does not involve human subjects research and is exempt from institutional review board approval.

## Population

The overall study population included all adults enrolled in CDM between 2017 and 2020. For annual incidence and prevalence, cohort eligibility was determined for each calendar year of study. Patients were required to be 18 years or older on Jan 1 of the calendar year under study and have continuous enrollment for the year of interest and 12-months prior. To determine an overarching period prevalence for 2017–2020 prevalence, patients were required to be at least 18 years of age on the date they met the baseline enrollment requirements for the 2017–2020 cohort. Study diagrams are provided in S1–S4 Figs.

The history of stroke subgroup required an inpatient medical claim with a primary diagnosis of stroke (ICD-10: H34.1, I63*, I64*) [41] between 1 October of the year prior to the year of interest and 30 September of the year of interest for the year over year cohorts, or between 1 October 2016 and 30 September 2020 for the 2017–2020 full period cohort. Additionally, the stroke event must have preceded the AF diagnosis (or occurred on the same day). Index date was defined as 1 January for the year over year cohorts, and as the first day that patients met the age and baseline enrollment requirements for the 2017–2020 full period cohort. This cohort was used as a proxy for a population that underwent systematic testing of AF within RWD as people who have an ischemic stroke are screened for AF with an ambulatory electrocardiography device such as a Holter monitor.

## Outcome and covariate definitions

In the CDM, race and ethnicity are combined in a single construct as Asian, Black, Hispanic, and non-Hispanic White. However, it is noted that race encompasses a broad set of social constructs influenced by societal, ancestral, and geographic factors, while ethnicity commonly refers to cultural background, often with a shared language or religion. As such, although there are diverse perspectives on preferred terminology, the emphasis of the study lies in understanding the impact of this covariate on health outcomes [42]. Other covariates of interest include the CDM-defined categories of education, occupation, net worth, family income, and enrollment. Missing or unknown were grouped together into one missing/unknown category per covariate. Demographic characteristics were assigned based on last value recorded prior to cohort entry.

AF events required at least 2 outpatient claims occurring at least 7 days to at most 365 days apart, or 1 inpatient claim containing ICD-10 diagnosis code I48* to be defined as having AF. Patients who met the outpatient definition were classified as having AF on the date of their second outpatient diagnosis. Patients who met the inpatient definition of AF were classified as having AF on the date of their inpatient admission [43].

Comorbid conditions of interest included obesity, hypertension, diabetes, cardiovascular disease, COPD, renal disease, and liver disease and were defined using previously validated ICD-10-CM diagnosis algorithms using inpatient and/or outpatient encounters. Comorbid conditions were assessed over the baseline period. Healthcare utilization was defined as the number of days with any medical or pharmacy claim in the year prior to index date, and patients were categorized by quartile. The $CHA_2DS_2$-VASc score was evaluated as a categorical variable based on the European Society of Cardiology guidelines categorizing the $CHA_2DS_2$-VASc score into three categories: 0,1, and $\geq 2$ [44].

## Statistical analysis

Distribution of demographic characteristics and comorbidities were assessed at time of cohort entry as mean (SD) or median (interquartile range [IQR]) for continuous variables and proportions (n, %) for categorical variables. Incidence and prevalence of AF were calculated annually for each year from 2017 to 2020, along with overall period prevalence in 2017–2020. Incident AF was defined as a new diagnosis of AF meeting the study algorithm definition. The incidence washout period was defined as the start of all available data until the 31$^{st}$ of December preceding the year of interest. Prevalent AF was defined as any diagnosis of AF meeting study criteria occurring during the time period of interest, without consideration of washout period. Incidence and prevalence calculations were performed for the overall cohort and within race and ethnicity strata. Prevalence estimates were further stratified by hypertension status, healthcare resource utilization (HCRU) quartile, and $CHA_2DS_2$-VASc score.

To assess the degree of potential over/under-ascertainment of AF among patients by racial group, the ratio between the prevalence of AF among highly monitored patients (i.e., those with a recent history of ischemic stroke) versus prevalence in the overall population was assessed. The equation used to determine the prevalence ratio within strata was:

$$\text{prevalence ratio} = \frac{\text{prevalence in stroke cohort}}{\text{prevalence in overall cohort}}$$

Additionally, the relative difference between the prevalence of AF in the NHW population is computed in relation to the prevalence of AF in other racial and ethnic populations. The equation used to determine the relative prevalence difference was:

$$\text{relative difference} = \frac{\text{prevalence in target group} - \text{prevalence in non-Hispanic or Latino White}}{\text{prevalence in non-Hispanic or Latino White}}$$

Relative prevalence differences are computed in both the overall and stroke population. Incidence and prevalence estimates were compared in different subgroups to explore potential ascertainment bias. Any observed differences in the overall cohort may be the result of differential testing and could point to potential ascertainment bias. Differences in the stroke cohort, following an acute event, are assumed to be closer to true differences, not originating from biases in healthcare delivery. All analyses were conducted using the Aetion Evidence Platform (2022).

## Results

### Overall population

The CDM population consisted of 42,392,287 patients from 2017 to 2020; 49% are male, 59% non-Hispanic or Latino White, 12.3% Hispanic or Latino, 9.9% Black or African American, and 4.9% are Asian. Approximately 39% of the population are aged 18–44 and nearly 25% of the population are aged 45–64 as shown in Table 1.

**Table 1. Demographic characteristics of the total and stroke population in the CDM dataset 2017–2020.**

| Characteristic | Total Population | Stroke Population |
|---|---|---|
| Number of patients | 42,392,287 | 597,300 |
| **Age (Categorical)** | | |
| ...< 18; n (%) | 17.3% | 0.3% |
| ...18–44; n (%) | 38.6% | 2.9% |
| ...45–64; n (%) | 24.7% | 19.5% |
| ...≥ 65; n (%) | 18.4% | 77.4% |
| ...Missing; n (%) | 1.0% | 0.0% |
| **Division (Geography)** | | |
| ...Missing/Unknown; n (%) | 1.9% | 0.1% |
| ...South Atlantic; n (%) | 22.6% | 27.5% |
| ...East South Central; n (%) | 4.7% | 5.0% |
| ...Middle Atlantic; n (%) | 7.2% | 9.8% |
| ...New England; n (%) | 3.1% | 4.0% |
| ...Mountain; n (%) | 9.6% | 8.2% |
| ...West North Central; n (%) | 9.1% | 6.7% |
| ...East North Central; n (%) | 14.1% | 12.8% |
| ...West South Central; n (%) | 16.5% | 14.4% |
| ...Pacific; n (%) | 11.2% | 11.5% |
| **Race** | | |
| ...Non-Hispanic or Latino White; n (%) | 59.0% | 65.2% |
| ...Asian; n (%) | 4.9% | 2.6% |
| ...Black or African American; n (%) | 9.9% | 15.8% |
| ...Hispanic or Latino; n (%) | 12.3% | 10.5% |
| ...Unknown/Missing; n (%) | 13.8% | 5.9% |
| **Sex** | | |
| ...Female; n (%) | 50.6% | 53.8% |
| ...Male; n (%) | 49.3% | 46.2% |
| ...Missing/Unknown; n (%) | 0.0% | 0.0% |

## Annual incidence

Approximately 7.5 to 8.6 million patients met the overall annual incidence cohort eligibility criteria each year. Of those, 107,864 to 142,507 patients met the stroke cohort eligibility criteria each year.

Table 2 shows that the annual AF incidence in the overall cohort ranged from 10.64 (10.57, 10.71) to 11.87 (11.79, 11.94) per 1,000 person-years (PY) between 2017–2020. In the cohort of patients with a recent stroke over the same period of time, the incidence of AF ranged from 66.06 (64.66, 67.45) to 71.95 (70.48, 73.43) per 1,000 PY. The incidence of AF in the overall cohort and stroke cohort was highly stable from 2017–2019, with year-over-year differences of less than 1.23 per 1,000 PY. In 2020, the incidence of AF among all patients decreased from the prior year's estimates by 0.5% and 10.4%, potentially due to COVID-19 pandemic-related changes to healthcare delivery and utilization. Similar AF incidence reductions were observed among patients with a recent ischemic stroke in 2020. Demographic and clinical characteristics did not differ year over year in the incident AF populations. The majority of incident AF patients were non-Hispanic or Latino White, had an income <$40,000 USD per year, had a history of hypertension, and had a mean age of ~74 years old. After stratifying by race, the annual AF incidence among all patients was highest for NHW patients (11.87–13.23), followed

**Table 2. Annual incidence of atrial fibrillation among all US adults vs. US adults with a recent history of ischemic stroke, 2017–2020.**

| | Number of patients at-risk All Adults | Incidence Rate (per 1000 PY) All Adults | Number of patients at-risk Recent Stroke | Incidence Rate (per 1000 PY) Recent Stroke |
|---|---|---|---|---|
| **2017** | 7,484,986 | 11.86 (11.79, 11.94) | 107,864 | 70.85 (69.23, 72.47) |
| . . .non-Hispanic or Latino White | 5,099,244 | 13.06 (12.97, 13.16) | 71,774 | 72.62 (70.61, 74.63) |
| . . .Black or African American | 766,249 | 12.07 (11.84, 12.31) | 17,466 | 61.67 (57.93, 65.42) |
| . . .Asian | 964,007 | 7.87 (7.70, 8.04) | 11,523 | 74.48 (69.39, 79.56) |
| . . .Hispanic or Latino | 387,580 | 4.84 (4.63, 5.04) | 3,060 | 62.28 (53.28, 71.28) |
| **2018** | 8,304,730 | 11.86 (11.79, 11.94) | 131,748 | 71.95 (70.48 73.43) |
| . . .non-Hispanic or Latino White | 5,614,284 | 13.06 (12.97, 13.16) | 88,092 | 75.63 (73.77, 77.48) |
| . . .Black or African American | 857,980 | 12.07 (11.84, 12.31) | 20,932 | 61.63 (58.21, 65.04) |
| . . .Asian | 1,094,047 | 7.87 (7.70, 8.04) | 14,130 | 64.38 (60.13, 68.64) |
| . . .Hispanic or Latino | 437,674 | 4.84 (4.63, 5.04) | 3,658 | 62.54 (54.30, 70.79) |
| **2019** | 8,575,224 | 11.87 (11.79, 11.94) | 142,507 | 69.49 (68.10, 70.89) |
| . . .non-Hispanic or Latino White | 5,760,419 | 13.23 (13.14, 13.33) | 94,923 | 73.90 (72.13, 75.66) |
| . . .Black or African American | 878,919 | 11.97 (11.74, 12.20) | 22,654 | 60.88 (57.62, 64.15) |
| . . .Asian | 1,100,384 | 7.73 (7.57, 7.90) | 14,962 | 59.27 (55.31, 63.23) |
| . . .Hispanic or Latino | 433,069 | 4.91 (4.70, 5.12) | 3,834 | 53.67 (46.23, 61.11) |
| **2020** | 8,579,998 | 10.64 (10.57, 10.71) | 134,349 | 66.06 (64.66, 67.45) |
| . . .non-Hispanic or Latino White | 5,690,704 | 11.87 (11.78, 11.96) | 88,740 | 69.47 (67.70, 71.23) |
| . . .Black or African American | 850,653 | 10.92 (10.69, 11.14) | 20,787 | 57.48 (54.18, 60.78) |
| . . .Asian | 1,070,978 | 7.40 (7.24, 7.57) | 13,820 | 60.22 (56.07, 64.37) |
| . . .Hispanic or Latino | 416,375 | 4.39 (4.19, 4.59) | 3,689 | 51.41 (44.00, 58.81) |

by Black or African American (10.92–12.07) and Asian (7.40–7.87) patients, and lowest for Hispanic or Latino patients (4.39–4.84). When assessed among patients with a recent ischemic stroke, Asian patients had the highest incidence of AF in 2017 (74.48) and the second highest incidence in 2018 and 2020 (64.38 and 60.22).

## Annual prevalence

Approximately 7.9 to 9.2 million patients met the eligibility criteria for the overall population for the estimation of yearly prevalence over 2017–2020, and 119,154 to 159,761 patients met the eligibility criteria for the population of patients with a recent history of ischemic stroke.

Between 2017–2020 the prevalence of AF ranged from 4.71% to 5.58% (Table 3). Among patients with a recent stroke, 2017–2020 annual prevalence of AF ranged from 25.88% to 27.27%. The prevalence of AF increased steadily from 2017–2019, with an overall change of 0.87%. In 2020, the prevalence of AF among all patients decreased from the prior year's estimates by 0.5% and 10.4% respectively, potentially due to COVID-19 pandemic-related changes to healthcare delivery and utilization. Similar reductions were observed in the prevalence of AF among patients with a recent ischemic stroke in 2020. Demographic and clinical characteristics of patients identified as having prevalent AF are reported in Table 4. The majority of prevalent AF patients were NHW, had an income <$40,000 USD per year, had a history of hypertension, and had a mean age of ~74 years old.

**Table 3. Prevalence ratio estimated from the annual prevalence of atrial fibrillation among all adults vs. adults with a recent ischemic stroke.**

| | Atrial Fibrillation Prevalence (%) All Adults | Atrial Fibrillation Prevalence (%) Recent Ischemic Stroke | Prevalence ratio* |
|---|---|---|---|
| **2017** | **4.71%** | **25.88%** | **5.49** |
| . . .non-Hispanic or Latino White | 5.32% | 28.22% | 5.30 |
| . . .Black or African American | 4.18% | 19.50% | 4.67 |
| . . .Asian | 2.80% | 21.37% | 7.63 |
| . . .Hispanic or Latino | 1.98% | 20.28% | 10.24 |
| **2018** | **5.22%** | **26.84%** | **5.14** |
| . . . non-Hispanic or Latino White | 5.94% | 29.18% | 4.91 |
| . . .Black or African American | 4.60% | 20.17% | 4.38 |
| . . .Asian | 3.08% | 22.26% | 7.23 |
| . . .Hispanic or Latino | 2.09% | 22.83% | 10.92 |
| **2019** | **5.58%** | **27.27%** | **4.89** |
| . . . non-Hispanic or Latino White | 6.39% | 29.80% | 4.66 |
| . . .Black or African American | 4.93% | 20.99% | 4.26 |
| . . .Asian | 3.30% | 22.43% | 6.80 |
| . . .Hispanic or Latino | 2.27% | 21.99% | 9.69 |
| **2020** | **5.55%** | **26.98%** | **4.86** |
| . . . non-Hispanic or Latino White | 6.40% | 29.35% | 4.59 |
| . . .Black or African American | 4.96% | 20.77% | 4.19 |
| . . .Asian | 3.40% | 22.92% | 6.74 |
| . . .Hispanic or Latino | 2.27% | 21.67% | 9.55 |

*Prevalence ratio is calculated as the prevalence among the population of patients with a recent stroke divided by the prevalence among the adult population, overall and within each stratification.

After stratifying by race, the prevalence of AF was highest for NHW patients (5.32% to 6.40%), followed by Black or African American (4.18% to 4.96%) and Asian (2.80% to 3.40%) patients, and lowest for Hispanic or Latino patients (1.98% to 2.27%), as shown in Table 3. In the recent ischemic stroke cohort, the relative prevalence differed, with Asian patients had the second highest prevalence of AF (after NHW patients) in 2017, 2019, and 2020.

## Prevalence ratio estimated from annual prevalence

To assess the degree of potential over/under-ascertainment of AF among patients by racial group, the prevalence ratio between the prevalence of AF among highly monitored patients (i.e., those with a recent history of ischemic stroke) versus prevalence in the overall population was assessed. The largest prevalence ratios were observed among Asian and Hispanic or Latino patients (6.74 to 7.63 for Asian patients; 9.55 to 10.92 for Hispanic or Latino patients—Table 3). For these patients, the prevalence of AF among the recent stroke population was approximately 7 to 10 times higher than the prevalence among the overall population (compared to a prevalence ratio approximately 5 times higher for NHW patients), suggesting that AF may be under-ascertained in the overall population for Asian and Hispanic or Latino patients.

## Period prevalence 2017–2020

The period prevalence for 2017–2020 was 5.09% in the overall population and 29.72% in the stroke population (Table 5). In both the overall population and the recent stroke population NHW patients had the highest period prevalence with 5.89% and 32.21% respectively. The

**Table 4. Selected patient characteristics for all adults with prevalent and incident atrial fibrillation annually, 2017–2020.**

| | Patients with Incident Atrial Fibrillation | | | | Patients with Prevalent Atrial Fibrillation | | | |
|---|---|---|---|---|---|---|---|---|
| | 2017 | 2018 | 2019 | 2020 | 2017 | 2018 | 2019 | 2020 |
| | N = 84, 251 | N = 97,947 | N = 101,158 | N = 91,071 | N = 371,872 | N = 460,124 | N = 510,264 | N = 510,348 |
| **Race Ethnicity** | | | | | | | | |
| . . .non-Hispanic or Latino White | 61,664 (73.19%) | 72,848 (74.37%) | 75,723 (74.86%) | 67,336 (73.94%) | 287,900 (77.42%) | 356,402 (77.46%) | 396,010 (77.61%) | 394,177 (77.24%) |
| . . .Black or African American | 8,852 (10.51%) | 10,295 (10.51%) | 10,458 (10.34%) | 9,263 (10.17%) | 33,517 (9.01%) | 41,536 (9.03%) | 45,870 (8.99%) | 44,798 (8.78%) |
| . . .Hispanic or Latino | 8,367 (9.93%) | 8,576 (8.76%) | 8,476 (8.38%) | 7,923 (8.70%) | 27,876 (7.50%) | 34,935 (7.59%) | 37,766 (7.40%) | 38,091 (7.46%) |
| . . .Asian | 1,846 (2.19%) | 2,112 (2.16%) | 2,121 (2.10%) | 1,829 (2.01%) | 7,881 (2.12%) | 9,368 (2.04%) | 10,091 (1.98%) | 9,748 (1.91%) |
| **Age** | | | | | | | | |
| . . .Mean (SD) | 73.68 (10.30) | 74.12 (10.31) | 74.12 (10.32) | 73.95 (10.36) | 75.28 (9.41) | 75.74 (9.29) | 75.91 (9.28) | 75.92 (9.29) |
| . . .Median [IQR] | 75.00 [68.00, 82.00] | 75.00 [69.00, 82.00] | 75.00 [69.00, 82.00] | 75.00 [68.00, 82.00] | 77.00 [70.00, 83.00] | 77.00 [70.00, 84.00] | 77.00 [71.00, 84.00] | 77.00 [71.00, 83.00] |
| **Gender** | | | | | | | | |
| . . .Female | 40,427 (48.0%) | 47,155 (48.1%) | 48,933 (48.4%) | 43,505 (47.8%) | 174,282 (46.9%) | 215,934 (46.9%) | 239,493 (46.9%) | 238,771 (46.8%) |
| . . .Male | 43,691 (51.9%) | 50,670 (51.7%) | 52,008 (51.4%) | 47,350 (52.0%) | 197,180 (53.0%) | 243,709 (53.0%) | 269,733 (52.9%) | 270,502 (53.0%) |
| . . .Missing /Unknown | 133 (0.2%) | 122 (0.1%) | 217 (0.2%) | 216 (0.2%) | 410 (0.1%) | 481 (0.1%) | 1,038 (0.2%) | 1,075 (0.2%) |
| **Geographic Region** | | | | | | | | |
| South Atlantic | 22,329 (26.5%) | 25,108 (25.6%) | 25,603 (25.3%) | 23,219 (25.5%) | 88,637 (23.8%) | 114,948 (25.0%) | 127,691 (25.0%) | 129,617 (25.4%) |
| East South Central | 2,676 (3.2%) | 5,370 (5.5%) | 5,415 (5.4%) | 3,754 (4.1%) | 11,909 (3.2%) | 22,615 (4.9%) | 26,240 (5.1%) | 20,003 (3.9%) |
| Middle Atlantic | 7,245 (8.6%) | 9,116 (9.3%) | 8,646 (8.5%) | 7,142 (7.8%) | 33,219 (8.9%) | 42,802 (9.3%) | 44,704 (8.8%) | 42,026 (8.2%) |
| New England | 2,804 (3.3%) | 3,605 (3.7%) | 4,973 (4.9%) | 4,361 (4.8%) | 14,598 (3.9%) | 18,882 (4.1%) | 26,749 (5.2%) | 27,189 (5.3%) |
| Mountain | 7,856 (9.3%) | 8,928 (9.1%) | 9,272 (9.2%) | 9,449 (10.4%) | 35,202 (9.5%) | 41,994 (9.1%) | 45,802 (9.0%) | 49,779 (9.8%) |
| West North Central | 6,971 (8.3%) | 7,708 (7.9%) | 7,520 (7.4%) | 7,217 (7.9%) | 32,381 (8.7%) | 37,359 (8.1%) | 37,788 (7.4%) | 39,450 (7.7%) |
| East North Central | 11,704 (13.9%) | 12,987 (13.3%) | 14,384 (14.2%) | 12,996 (14.3%) | 56,602 (15.2%) | 64,918 (14.1%) | 74,939 (14.7%) | 76,956 (15.1%) |
| West South Central | 10,173 (12.1%) | 11,734 (12.0%) | 12,305 (12.2%) | 11,896 (13.1%) | 41,170 (11.1%) | 49,632 (10.8%) | 55,937 (11.0%) | 58,786 (11.5%) |
| Pacific | 12,302 (14.6%) | 13,169 (13.4%) | 12,740 (12.6%) | 10,768 (11.8%) | 57,596 (15.5%) | 66,215 (14.4%) | 69,137 (13.5%) | 65,268 (12.8%) |
| Missing Unknown | 191 (0.2%) | 222 (0.2%) | 300 (0.3%) | 269 (0.3%) | 558 (0.2%) | 759 (0.2%) | 1,277 (0.3%) | 1,274 (0.2%) |
| **Household Income** | | | | | | | | |
| ≥ $100 | 15,046 (17.9%) | 18,398 (18.8%) | 19,737 (19.5%) | 17,811 (19.6%) | 69,993 (18.8%) | 89,646 (19.5%) | 101,598 (19.9%) | 101,646 (19.9%) |
| $75 - $99 | 11,693 (13.9%) | 14,172 (14.5%) | 15,322 (15.1%) | 13,571 (14.9%) | 54,517 (14.7%) | 70,234 (15.3%) | 80,154 (15.7%) | 80,884 (15.8%) |
| $60 - $74 | 9,404 (11.2%) | 11,376 (11.6%) | 11,825 (11.7%) | 10,471 (11.5%) | 42,391 (11.4%) | 54,368 (11.8%) | 61,162 (12.0%) | 60,977 (11.9%) |
| $50 - $59 | 7,803 (9.3%) | 9,243 (9.4%) | 9,681 (9.6%) | 8,735 (9.6%) | 34,863 (9.4%) | 44,062 (9.6%) | 49,752 (9.8%) | 49,901 (9.8%) |
| $40 - $49 | 7,570 (9.0%) | 8,771 (9.0%) | 9,258 (9.2%) | 8,194 (9.0%) | 33,081 (8.9%) | 41,034 (8.9%) | 46,190 (9.1%) | 45,764 (9.0%) |
| < $40 | 26,240 (31.1%) | 29,562 (30.2%) | 30,271 (29.9%) | 27,383 (30.1%) | 111,185 (29.9%) | 134,707 (29.3%) | 148,815 (29.2%) | 147,899 (29.0%) |
| Missing/Unknown | 6,495 (7.7%) | 6,425 (6.6%) | 5,064 (5.0%) | 4,906 (5.4%) | 25,842 (6.9%) | 26,073 (5.7%) | 22,593 (4.4%) | 23,277 (4.6%) |
| **Education Level** | | | | | | | | |
| Bachelor Degree Plus | 12,165 (14.4%) | 14,166 (14.5%) | 15,149 (15.0%) | 13,015 (14.3%) | 59,074 (15.9%) | 72,943 (15.9%) | 80,322 (15.7%) | 77,867 (15.3%) |

*(Continued)*

**Table 4.** (Continued)

| | Patients with Incident Atrial Fibrillation | | | | Patients with Prevalent Atrial Fibrillation | | | |
|---|---|---|---|---|---|---|---|---|
| | **2017** | **2018** | **2019** | **2020** | **2017** | **2018** | **2019** | **2020** |
| | **N = 84, 251** | **N = 97,947** | **N = 101,158** | **N = 91,071** | **N = 371,872** | **N = 460,124** | **N = 510,264** | **N = 510,348** |
| Less than Bachelor Degree | 45,762 (54.3%) | 54,088 (55.2%) | 55,434 (54.8%) | 49,712 (54.6%) | 208,807 (56.2%) | 259,618 (56.4%) | 287,416 (56.3%) | 287,045 (56.2%) |
| High School Diploma | 23,388 (27.8%) | 26,199 (26.7%) | 26,726 (26.4%) | 24,081 (26.4%) | 92,252 (24.8%) | 113,060 (24.6%) | 125,354 (24.6%) | 124,672 (24.4%) |
| Less than 12th Grade | 338 (0.4%) | 367 (0.4%) | 360 (0.4%) | 346 (0.4%) | 1,294 (0.3%) | 1,419 (0.3%) | 1,502 (0.3%) | 1,519 (0.3%) |
| Missing/Unknown | 2,598 (3.1%) | 3,127 (3.2%) | 3,489 (3.4%) | 3,917 (4.3%) | 10,445 (2.8%) | 13,084 (2.8%) | 15,670 (3.1%) | 19,245 (3.8%) |
| **Number of days with healthcare interactions, median [IQR]** | 15.00 [7.00, 29.00] | 16.00 [9.00, 31.00] | 17.00 [9.00, 31.00] | 17.00 [9.00, 32.00] | 24.00 [14.00, 42.00] | 25.00 [14.00, 43.00] | 25.00 [14.00, 44.00] | 25.00 [14.00, 44.00] |
| **Comorbidities** | | | | | | | | |
| Ischemic stroke | 3,550 (4.2%) | 4,614 (4.7%) | 4,994 (4.9%) | 4,499 (4.9%) | 24,934 (6.7%) | 32,864 (7.1%) | 37,177 (7.3%) | 37,266 (7.3%) |
| Obesity | 12,508 (14.8%) | 16,680 (17.0%) | 19,244 (19.0%) | 18,645 (20.5%) | 69,535 (18.7%) | 95,321 (20.7%) | 115,804 (22.7%) | 125,360 (24.6%) |
| Hypertension | 61,100 (72.5%) | 75,776 (77.4%) | 78,197 (77.3%) | 70,287 (77.2%) | 311,406 (83.7%) | 395,212 (85.9%) | 440,565 (86.3%) | 442,249 (86.7%) |
| Diabetes | 27,167 (32.2%) | 32,723 (33.4%) | 33,775 (33.4%) | 30,369 (33.3%) | 128,868 (34.7%) | 161,175 (35.0%) | 178,179 (34.9%) | 179,395 (35.2%) |
| Cardiovascular Disease | 15,975 (19.0%) | 19,814 (20.2%) | 20,516 (20.3%) | 18,587 (20.4%) | 131,529 (35.4%) | 168,714 (36.7%) | 190,175 (37.3%) | 194,709 (38.2%) |
| Chronic Obstructive Pulmonary Disease (COPD) | 14,599 (17.3%) | 17,824 (18.2%) | 18,234 (18.0%) | 15,912 (17.5%) | 78,103 (21.0%) | 98,580 (21.4%) | 109,139 (21.4%) | 109,512 (21.5%) |
| Renal diseases | 16,388 (19.5%) | 20,728 (21.2%) | 21,921 (21.7%) | 20,881 (22.9%) | 95,266 (25.6%) | 122,790 (26.7%) | 140,792 (27.6%) | 149,319 (29.3%) |
| Liver Disease | 3,166 (3.8%) | 3,999 (4.1%) | 4,522 (4.5%) | 4,327 (4.8%) | 15,195 (4.1%) | 20,413 (4.4%) | 24,346 (4.8%) | 26,246 (5.1%) |
| **CHA$_2$DS$_2$-VASc score** | | | | | | | | |
| 0 | 4,852 (5.8%) | 5,107 (5.2%) | 5,087 (5.0%) | 4,471 (4.9%) | 13,612 (3.7%) | 13,627 (3.0%) | 13,546 (2.7%) | 12,388 (2.4%) |
| 1 | 11,212 (13.3%) | 11,430 (11.7%) | 11,273 (11.1%) | 10,121 (11.1%) | 35,107 (9.4%) | 36,084 (7.8%) | 35,831 (7.0%) | 33,499 (6.6%) |
| 2 | 20,468 (24.3%) | 20,356 (20.8%) | 19,738 (19.5%) | 17,246 (18.9%) | 71,087 (19.1%) | 73,513 (16.0%) | 73,087 (14.3%) | 67,473 (13.2%) |
| 3 | 21,934 (26.0%) | 24,010 (24.5%) | 23,203 (22.9%) | 19,984 (21.9%) | 88,767 (23.9%) | 98,500 (21.4%) | 98,593 (19.3%) | 91,395 (17.9%) |
| 4 | 12,576 (14.9%) | 16,264 (16.6%) | 16,822 (16.6%) | 14,688 (16.1%) | 67,580 (18.2%) | 83,335 (18.1%) | 90,286 (17.7%) | 87,074 (17.1%) |
| 5 | 7,017 (8.3%) | 9,963 (10.2%) | 11,229 (11.1%) | 10,632 (11.7%) | 46,364 (12.5%) | 66,393 (14.4%) | 77,804 (15.2%) | 80,161 (15.7%) |
| ≥6 | 6,192 (7.3%) | 10,817 (11.0%) | 13,806 (13.6%) | 13,929 (15.3%) | 49,355 (13.3%) | 88,672 (19.3%) | 121,117 (23.7%) | 138,358 (27.1%) |

Asian patients had the lowest prevalence in the overall population with a prevalence of 2.08% while the Black or African American patient population had the lowest prevalence in the stroke patient population at 24.26%.

## Prevalence ratio estimated from period prevalence 2017–2020

The largest prevalence ratios were observed among Asian and Hispanic or Latino patients (7.88 and 11.75 respectively–Table 5), for whom the prevalence of AF among the recent stroke population was approximately 7 to 11 times higher than the prevalence among the overall population (compared to a prevalence ratio approximately 5 times higher for NHW patients),

**Table 5. Prevalence ratio estimated from the period prevalence of atrial fibrillation among all adults vs. adults with a recent ischemic stroke in 2017–2020.**

| | Atrial Fibrillation Period Prevalence (%) All Adults | Atrial Fibrillation Period Prevalence (%) Recent Ischemic Stroke | Prevalence ratio* |
|---|---|---|---|
| **Overall** | 5.09% | 29.72% | 5.84 |
| . . . non-Hispanic or Latino White | 5.89% | 32.21% | 5.47 |
| . . . Black or African American | 4.93% | 24.26% | 4.92 |
| . . . Asian | 2.08% | 24.45% | 11.75 |
| . . . Hispanic or Latino | 3.17% | 24.99% | 7.88 |
| **No Hypertension** | | | |
| Overall | 1.06% | 17.00% | 16.04 |
| . . . non-Hispanic or Latino White | 1.29% | 17.93% | 13.90 |
| . . . Black or African American | 0.79% | 14.36% | 18.18 |
| . . . Asian | 0.40% | 13.08% | 32.70 |
| . . . Hispanic or Latino | 0.68% | 15.68% | 23.06 |
| **Hypertension** | | | |
| Overall | 11.82% | 31.91% | 2.70 |
| . . . non-Hispanic or Latino White | 13.24% | 34.84% | 2.63 |
| . . . Black or African American | 9.50% | 25.27% | 2.66 |
| . . . Asian | 7.05% | 26.88% | 3.81 |
| . . . Hispanic or Latino | 8.20% | 26.73% | 3.26 |
| **HCRU Quartile 1** | | | |
| Overall | 0.95% | 22.54% | 23.73 |
| . . . non-Hispanic or Latino White | 1.15% | 23.75% | 20.65 |
| . . . Black or African American | 0.89% | 18.78% | 21.10 |
| . . . Asian | 0.37% | 23.52% | 63.57 |
| . . . Hispanic or Latino | 0.84% | 17.44% | 20.76 |
| **HCRU Quartile 2** | | | |
| Overall | 2.65% | 22.30% | 8.42 |
| . . . non-Hispanic or Latino White | 2.99% | 24.26% | 8.11 |
| . . . Black or African American | 2.52% | 18.18% | 7.21 |
| . . . Asian | 1.35% | 18.57% | 13.76 |
| . . . Hispanic or Latino | 1.75% | 18.58% | 10.62 |
| **HCRU Quartile 3** | | | |
| Overall | 5.53% | 25.22% | 4.56 |
| . . . non-Hispanic or Latino White | 6.11% | 27.37% | 4.48 |
| . . . Black or African American | 5.07% | 20.36% | 4.02 |
| . . . Asian | 3.28% | 21.78% | 6.64 |
| . . . Hispanic or Latino | 3.80% | 20.93% | 5.51 |
| **HCRU Quartile 4** | | | |
| Overall | 14.59% | 34.98% | 2.40 |
| . . . non-Hispanic or Latino White | 15.65% | 37.86% | 2.42 |
| . . . Black or African American | 13.73% | 28.43% | 2.07 |
| . . . Asian | 8.65% | 30.55% | 3.53 |
| . . . Hispanic or Latino | 10.81% | 29.29% | 2.71 |

(*Continued*)

**Table 5.** (Continued)

| | Atrial Fibrillation Period Prevalence (%) All Adults | Atrial Fibrillation Period Prevalence (%) Recent Ischemic Stroke | Prevalence ratio* |
|---|---|---|---|
| **CHA$_2$DS$_2$-VASc = 0** | | | |
| Overall | 0.79% | 14.02% | 17.75 |
| . . . non-Hispanic or Latino White | 0.99% | 15.20% | 15.35 |
| . . . Black or African American | 0.68% | 11.65% | 17.13 |
| . . . Asian | 0.25% | 10.90% | 43.60 |
| . . . Hispanic or Latino | 0.38% | 10.28% | 27.05 |
| **CHA$_2$DS$_2$-VASc = 1** | | | |
| Overall | 1.54% | 18.95% | 12.31 |
| . . . non-Hispanic or Latino White | 1.86% | 20.59% | 11.07 |
| . . . Black or African American | 1.31% | 15.74% | 12.02 |
| . . . Asian | 0.59% | 14.26% | 24.17 |
| . . . Hispanic or Latino | 0.82% | 14.71% | 17.94 |
| **CHA$_2$DS$_2$-VASc $\geq$ 2** | | | |
| Overall | 13.91% | 39.10% | 2.81 |
| . . . non-Hispanic or Latino White | 15.54% | 43.12% | 2.77 |
| . . . Black or African American | 11.89% | 30.43% | 2.56 |
| . . . Asian | 7.97% | 32.30% | 4.05 |
| . . . Hispanic or Latino | 9.86% | 32.99% | 3.35 |

*Prevalence ratio is calculated as the prevalence among the population of patients with a recent stroke divided by the prevalence among all adult population, overall and within each stratification.

Abbreviations: HCRU Healthcare Resource Utilization

suggesting that AF may be under-ascertained in the overall population for Asian and Hispanic or Latino patients.

The largest prevalence ratio was seen in the no hypertension subgroup with the lowest prevalence ratio in NHW patients (13.90) and the highest prevalence ratio in the Asian (32.70) and Hispanic or Latino (23.06) patients. This is in sharp contrast to the patterns seen in the hypertension subgroup where the prevalence ratio across racial and ethnic groups ranged from 2.63 among NHW patients to 3.81 among Asian patients. In the HCRU quartile stratified results, the largest prevalence ratios was seen in the first HCRU quartile with a range of 20.65 among NHW patients to 63.57 among Asian patients while the lowest prevalence ratios was seen in the fourth HCRU quartile where the prevalence ratio across racial and ethnic groups ranged from 2.07 among Black or African American patients to 3.53 among Asian patients.

In the results stratified by CHA$_2$DS$_2$-VASc score categories the largest overall prevalence ratio was in the CHA$_2$DS$_2$-VASc score of 0 group (17.75) where the lowest prevalence ratio was among NHW patients (15.35) and the highest prevalence ratio was observed among Asian and Hispanic or Latino patients (43.60 and 27.05 respectively). The prevalence ratio decreases as the CHA$_2$DS$_2$-VASc score increases with the lowest prevalence ratio seen in the CHA$_2$DS$_2$-VASc score $\geq$2 subgroup (range 2.56 among Black or African American patients to 4.05 among Asian patients).

### Relative difference in period prevalence 2017–2020

To allow for a comparison of the results with prior observational studies, the relative difference of 2017–2020 period prevalence in NHW population is computed in relation to the other racial

**Table 6. An overview of relative period prevalence difference among all adults vs. adults with a recent ischemic stroke in 2017–2020.**

| | AF Period Prevalence Relative Difference versus NHW population (%) | |
|---|---|---|
| | **All Adults** | **Recent Ischemic Stroke** |
| . . .Asian | -46.2% | -22.4% |
| . . .Black or African American | -16.3% | -24.7% |
| . . .Hispanic or Latino | -64.6% | -24.1% |

and ethnic populations in Table 6. Among the overall population, the greatest magnitude of relative difference was seen in the Hispanic or Latino population (64.6%), followed by the Asian (46.2%) and the Black or African American (16.3%) population. Relative differences in period prevalence change in the stroke population (respectively, 24.7%, 24.1% and 22.4% for Black or African American, Hispanic, and Asian populations) and relative differences across racial and ethnic populations are more similar compared to the overall population.

## Discussion

This study explores prevalence and incidence of AF in two cohorts: one cohort reflecting standard clinical practice in the US, and the other mimicking a population subjected to AF monitoring or screening. Unlike previous observational research, which tends to focus on either clinical practice or monitoring separately, this study investigates both settings within the same population. This approach is key to assessing the impact of ascertainment bias quantitatively across subgroups.

The investigation is performed on a large-scale claims database (CDM) which represents the demographics of the United States across various geographical regions and insurance categories. In contract, other observational studies often target smaller, and regionally or demographically limited populations. Additionally, the study's scope encompasses an analysis of Asian and Hispanic or Latino subgroups, which historically have been less studied in US observational AF studies.

The findings of this study substantiate prior hypotheses regarding racial and ethnic variations in AF prevalence. Comparing, for example, annual and period AF prevalence in the CDM population with prior observational studies, we identified a consistent pattern. In both cohorts, the Black or African American, Hispanic or Latino, and Asian population have a lower AF prevalence compared to the NHW population. However, the observed differences are assumed to be attenuated by ascertainment bias and are expected to be less pronounced than initially estimated, particularly in the case of Asian and Hispanic or Latino populations.

Our findings show the greatest prevalence ratios across the overall and stroke cohort were observed among Asian and Hispanic or Latino patients, suggesting that AF may be under-ascertained in the overall population for these patients. In the social determinants of health (SDOH) subgroups, the greatest prevalence ratios was observed in the 'no hypertension' subgroup, the first HCRU quartile and the group with $CHA_2DS_2$-VASc score of 0 with a consistent pattern of the lowest prevalence ratio in NHW patients and the highest prevalence ratio in the Asian and Hispanic or Latino patients. The most significant disparity in ascertainment of AF is thus observed among Asian and Hispanic or Latino patients, overall and within subgroups characterized by good overall health (HCRU quartile 1) and a minimal presence of AF risk factors. For NHW and Black or African American populations with comparable health statuses, the gap in AF ascertainment is less pronounced compared to Asian and Hispanic or Latino patients. However, it is essential to recognize that even within these populations, disparities in AF diagnosis and detection may still exist, warranting further investigation.

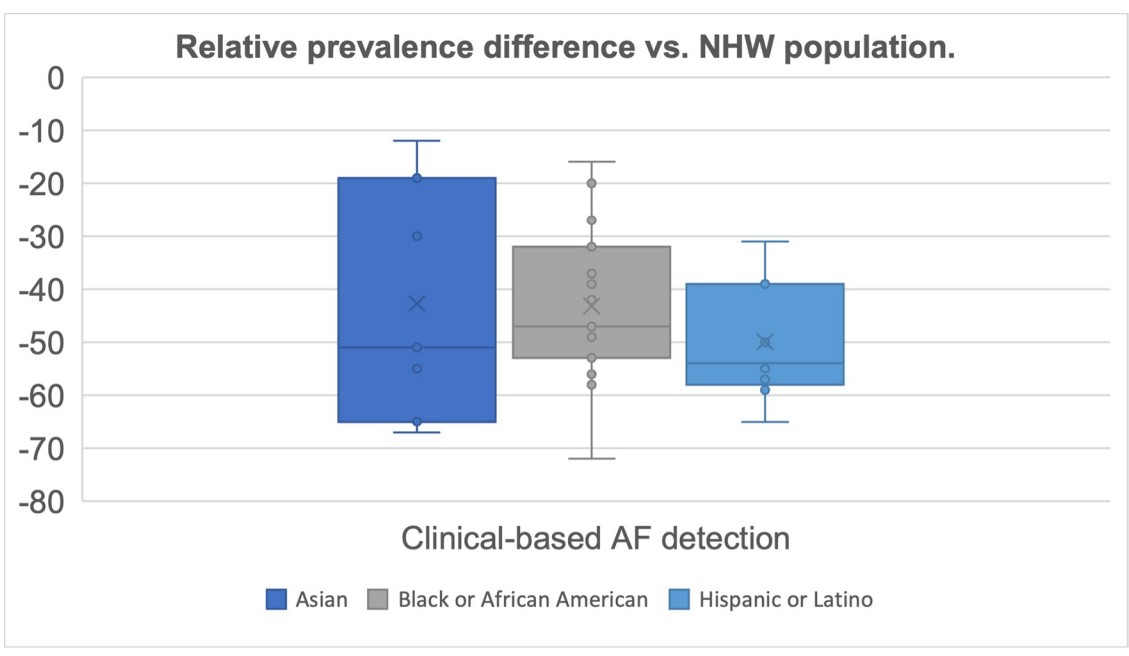

**Fig 1. Overview of relative AF prevalence difference of Asian, Black or African American, and Hispanic or Latino subgroup versus the NHW population across observational studies that used clinical detection methods.**

To facilitate comparisons between the various studies and our results, Fig 1 presents the relative prevalence difference of the NHW subgroup in relation to other racial and ethnic subgroups. Prevalence estimates across racial and ethnic subgroups from the different observational studies that used clinical detection methods are described in S2 Table. The focus is on prevalence results as incidence rates are not frequently reported in prior studies. It is important to note that variations in the definitions of racial/ethnic groups exist, as well as differences in the patient populations included across the observational studies. Most observational studies target the overall adult population, yet some target populations with prior comorbidities, specific age groups or ethnicities, resulting in a broad range of results [26, 28, 31, 32].

Comparing across studies, the Black or African American, Hispanic or Latino and Asian AF prevalence is estimated to be respectively 44%, 50% and 43% lower than the AF prevalence for the NHW subgroup (average reported across respectively 16, 9 and 7 studies). For the Asian and Hispanic or Latino population, the findings of the overall adult cohort from this study, which relies on clinical based detection, align with the findings of prior studies, and highlight similar differences in prevalence of 46.2% and 64.6% respectively. The prevalence difference of the Black or African American population is estimated lower than what is observed in prior studies at 16.3%. Although prevalence differences vary driven by study design and population, the results indicate that minority subgroups consistently have a lower AF prevalence than NHW subgroups.

To better understand and address the AF paradox, several observational studies have been conducted, employing monitoring-based methods for the diagnosis of AF. An overview of AF prevalence across these studies is provided in S3 Table. Comparing across studies, the Black or African American, Hispanic or Latino and Asian AF prevalence is estimated to be respectively, 23%, 3% and 27% lower than the AF prevalence for the NHW subgroup (average reported across respectively number of studies 3, 1 and 1 studies). The variability in detection of AF by

diagnosis method underscores potential disparities in observational data, as studies where patients were systematically screened for AF using monitoring-based methods, such as MESA [25] and a pacemaker-based study [33], showed lower prevalence differences among subgroups. Not all studies address all races and ethnicities, and significant differences exist across studies. The findings of the stroke cohort from this study, which assumes close monitoring of the patients, are directionally similar, with a prevalence difference of respectively, 24.7%, 24.1% and 22.4.% for Black or African American, Hispanic or Latino, and Asian populations. While the prevalence gap for the Asian and Hispanic or Latino population decreases by more than 50%, it increases for the Black or African American population. This could be ascribed to differences in ischemic stroke risk factors, etiology and outcomes in Black or African American patients [45].

The number of studies focused on racial/ethnic differences in AF monitoring is limited, especially for Asian and Hispanic or Latino populations. When comparing the relative prevalence differences of the NHW subgroup with Black or African American, Hispanic or Latino, and Asian subgroups together with the results from past observational studies, a difference is observed based on the method of AF detection employed in the studies. Specifically, results that relied on clinical-based detection methods, report higher relative prevalence differences than results that relied on monitoring-based detection methods. Studies utilizing continuous monitoring procedures or events that assume monitoring tend to reveal lower relative prevalence differences among these subgroups. This would suggest that when standard clinical assessments are used for AF diagnosis, differences in prevalence among racial and ethnic subgroups are overestimated due to ascertainment bias.

Despite the method used for AF diagnosis, significant differences in prevalence remain, indicating that factors other than ascertainment bias are also likely contributors. We conclude that while ascertainment bias is unlikely to explain all differences in AF prevalence observed by race and ethnicity, current clinical estimates, whether due to access to care or testing approaches, are less accurate in reflecting the 'true' burden of AF within Asian and Hispanic or Latino patient populations. Factors that can contribute to ascertainment bias of AF in these racial and ethnic groups include language barriers, socioeconomic factors, cultural barriers, and differences in healthcare-seeking behavior. These findings underscore the importance of not only the diagnostic methods employed when assessing AF prevalence, but also the importance of patient education, culturally competent care, and improved and equal access to treatment and care including appropriate screening and diagnostic protocols.

There is an immediate opportunity to apply these findings in the clinical trial setting. When using RWD in designing a trial, it is imperative we proactively account for data biases arising from known health disparities. For example, when establishing evidence-based diversity enrollment targets, it is critical to exercise caution and address potential ascertainment bias that can differentially affect minority groups. In the specific context of AF RCTs, ascertainment bias can introduce disparities in the identification, diagnosis, and enrollment of individuals from different racial and ethnic backgrounds. Acknowledging and addressing ascertainment challenges can improve diverse enrollment in AF RCTs. In turn, this will contribute to generating robust evidence for populations affected by AF.

## Limitations

This study examines ascertainment bias in AF, and the study design controls for such bias, by contrasting the overall population with a cohort of patients with a recent ischemic stroke, for which AF monitoring is part of stroke guidelines [46]. The study relies on claims data to characterize differences, as opposed to observational studies that rely on active monitoring-based

diagnosis methods for AF to assess the impact of ascertainment bias in AF differences across racial and ethnic groups.

There are several potential limitations to consider when using real-world administrative data to assess prevalence and incidence for underrepresented groups. This study relies on the use of secondary data collected for administrative, and not research, purposes. Results from this study depend on the accuracy of diagnostic codes used in the diagnosis of a health state. There is the potential for miscoding in the atrial fibrillation and ischemic stroke events and subsequent risk of misclassification bias in patient records due to provider coding patterns (e.g., using diagnosis codes to indicate rule-out criterion) or incorrect coding (e.g., data entry errors), which may lead to misclassification of diagnoses or patient characteristics. The algorithms identified for the current study have been previously validated with a focus on precision [41, 43]. However, the calendar time components used in some of the algorithms introduce the potential for undercounting people with the outcome of interest, particularly if there are issues with continuity of care over the course of the 2-year timeframe. Using a requirement of continuous enrollment in the baseline period could result in selection bias and underreporting of events. Gaps in coverage could also contribute to underreporting. These limitations could result in an underestimation of the incidence and prevalence of these outcomes.

Additionally, the use of derived race and ethnicity information in the CDM is also likely to be less accurate compared to self-reported data. Prior research has shown that NHW and Black or African American individuals are more likely to be correctly identified in healthcare data, while individuals who identify as Asian, Hispanic or Latino, or multi-racial backgrounds are often inaccurately categorized as White [18]. As such, misclassification of demographics such as race, ethnicity, and other factors used in the analysis cannot be ruled out, nor is it possible to make assumptions on AF prevalence and incidence patterns for patients in the missing/unknown category.

Another limitation of this study is the factors contributing to health disparities in atrial fibrillation further downstream from diagnosis. These include disparities in access to care, disparities in diagnosis patterns, and disparities in health coverage. The data used for this study will not be able to assess the contributions of these downstream factors to the incidence and prevalence of the outcomes. However, the data presented from this study will reflect the real-world experiences of insured patients in the US.

Finally, in the study design, it is presumed that stroke guidelines are closely adhered to, especially the active monitoring of patients following an ischemic stroke event. Although the use of monitoring procedure codes was also considered in the cohort design, their implementation was excluded as limited validation is available for these monitoring procedure codes. It is also important to consider that, as our second cohort only consists of patients with a past ischemic stroke event, it might not be representative of a general population. However, to support the generalizability of our study, we included the patient characteristics of both cohorts in Table 1 and investigated the etiology of ischemic strokes. The NHW have a significantly greater proportion of cardioembolic stroke than the Hispanic or Latino and Black or African American population, which is the leading stroke subtype caused by atrial fibrillation and/or flutter. For other stroke subtypes associated with AF, such as cryptogenic stroke, proportions were not significantly different among the 3 racial/ethnic groups [45]. As the proportion of AF associated stroke subtypes is largest for the NHW subgroup, the reduction in AF prevalence gap between subgroups is less likely to be driven by differences in ischemic stroke etiology.

## Conclusion

Racial and ethnic minority groups face significant disparities in healthcare, including under-representation in clinical trials. Underrepresentation can hinder the application of trial

findings to these populations. The FDA has published guidelines to increase representation in trials and reduce racial and ethnic health disparities by setting enrollment targets for underrepresented groups. Real-world data (RWD) is valuable in estimating disease incidence and prevalence among racial and ethnic subgroups to inform these enrollment targets.

However, assessing racial and ethnic differences in RWD requires distinguishing true disease differences from biases originating from healthcare delivery, such as ascertainment bias. This study utilizes US claims RWD to estimate incidence and prevalence of atrial fibrillation (AF) within racial and ethnic groups, considering both the overall adult population and a highly monitored population to evaluate ascertainment bias.

The study findings align with previous observational studies, revealing lower incidence and prevalence rates of AF in US racial/ethnic minority groups. However, a key element influencing the reported prevalence differences is the choice of AF diagnostic methods. Specifically, prevalence estimates derived from routine clinical-based detection methods exhibit higher relative prevalence differences compared to monitoring-based detection methods, particularly among healthy Asian and Hispanic or Latino subgroups.

These findings emphasize the importance of considering the diagnostic method when assessing AF prevalence. Similarly, addressing data biases linked to health disparities, is crucial in the use of real-world evidence to define recruitment targets for underrepresented groups.

## Supporting information

**S1 Table. Overview of atrial fibrillation observational studies.**
(DOCX)

**S2 Table. Overview of atrial fibrillation prevalence by racial and ethnic groups across observational studies (clinical-based atrial fibrillation detection).**
(DOCX)

**S3 Table. Overview of atrial fibrillation prevalence by racial and ethnic groups across observational studies (monitoring-based atrial fibrillation detection).**
(DOCX)

**S1 Fig. Incidence of atrial fibrillation among all US adults.**
(JPG)

**S2 Fig. Incidence of atrial fibrillation among US adults with recent ischemic stroke hospitalization.**
(JPG)

**S3 Fig. Prevalence of atrial fibrillation among all US adults.**
(JPG)

**S4 Fig. Prevalence of atrial fibrillation among US adults with recent ischemic stroke hospitalization.**
(JPG)

## Acknowledgments

The authors thank Preston Dunnmon and Ilker Oztop for their valuable contributions.

## Author Contributions

**Conceptualization:** Lars Hulstaert, Amelia Boehme, Kaitlin Hood, Jennifer Hayden, Clark Jackson, Astra Toyip, Hans Verstraete, Yu Mao, Khaled Sarsour.

**Data curation:** Lars Hulstaert, Amelia Boehme, Kaitlin Hood, Jennifer Hayden, Clark Jackson, Astra Toyip, Khaled Sarsour.

**Formal analysis:** Lars Hulstaert, Amelia Boehme, Kaitlin Hood, Jennifer Hayden, Clark Jackson, Astra Toyip, Khaled Sarsour.

**Investigation:** Lars Hulstaert, Amelia Boehme, Kaitlin Hood, Jennifer Hayden, Clark Jackson, Astra Toyip, Khaled Sarsour.

**Methodology:** Lars Hulstaert, Amelia Boehme, Kaitlin Hood, Jennifer Hayden, Clark Jackson, Astra Toyip, Hans Verstraete, Yu Mao, Khaled Sarsour.

**Visualization:** Lars Hulstaert, Amelia Boehme, Kaitlin Hood, Jennifer Hayden, Clark Jackson, Astra Toyip, Khaled Sarsour.

**Writing – original draft:** Lars Hulstaert.

**Writing – review & editing:** Lars Hulstaert, Amelia Boehme, Kaitlin Hood, Jennifer Hayden, Clark Jackson, Astra Toyip, Hans Verstraete, Yu Mao, Khaled Sarsour.

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
