## [Decision Letter · Decision Letter 0]

27 Feb 2024

PONE-D-23-40160Assessing ascertainment bias in atrial fibrillation across US minority groups.PLOS ONE

Dear Dr. Hulstaert,

Thank you for submitting your manuscript to PLOS ONE. After careful consideration, we feel that it has merit but does not fully meet PLOS ONE’s publication criteria as it currently stands. Therefore, we invite you to submit a revised version of the manuscript that addresses the points raised during the review process.

Here are my comments: While the study focused on investigating the presence of ascertainment bias, the study's design lends itself to ascertainment bias; in that those patients with a recent stroke would have increased focus on recognizing AF. I encourage the authors to discuss this oddity and include language or evidence to support the stance that claims data represents real-world data.

Line 111: Ref 27 is repeated. Please edit.

Again, in Line 111, I suggest using “blacks” instead of “African Americans” as “blacks” is more encompassing and would be in keeping with the style of the CMD. Alternatively, provide a working definition of your preferred terms.

Line 153: “The patient’s name and geography are...” Are the authors describing the data vendor’s treatment of the claims data or their manipulation of it? Please clarify as the latter situation may be misconstrued as a HIPAA breach. The statement on Line 156 claiming de-identified data must be reconciled with the previous quote.

Line 170: Please note that including ICD-10 codes such as I64.x (which could include hemorrhagic strokes) will reduce the specificity of your inclusion. One would not routinely monitor for AF in hemorrhagic strokes. Please clarify or include as a limitation in the discussion.

The expression shown on Line 226 is the prevalence ratio (by definition), but it was called “magnitude of difference”, which would invoke the absolute difference. Since this expression is used throughout the paper, it constitutes a major flaw.

We look forward to receiving your revised manuscript.

Kind regards,

Daniel Antwi-Amoabeng, MD, MSc

Academic Editor

PLOS ONE

"The authors received no specific funding for this work. LH, KH, JH, HV, YM & KS are full-time salaried employees of Janssen Research & Development, a pharmaceutical company of Johnson & Johnson. AB, CJ & AT are full-time salaried employees of Aetion, Inc. The specific roles of these authors are articulated in the ‘author contributions’ section. The funders had no role in study design, data collection and analysis, decision to publish, or preparation of the manuscript."

"LH, KH, JH, HV, YM & KS are employees of Janssen Research and Development, a unit of Johnson and Johnson family of companies. The work on this study was part of their employment.

AB, CJ & AT & are paid employees of and shareholders in Aetion, Inc., a company that makes software for the analysis of real-world data. 

This does not alter our adherence to PLOS One policies on sharing data and material." 

5. We note that you have indicated that there are restrictions to data sharing for this study. PLOS only allows data to be available upon request if there are legal or ethical restrictions on sharing data publicly. For more information on unacceptable data access restrictions, please see http://journals.plos.org/plosone/s/data-availability#loc-unacceptable-data-access-restrictions. 

Reviewers' comments:

Reviewer's Responses to Questions

**Comments to the Author**

1. Is the manuscript technically sound, and do the data support the conclusions?

Reviewer #1: Partly

Reviewer #2: Yes

2. Has the statistical analysis been performed appropriately and rigorously? 

Reviewer #1: Yes

Reviewer #2: Yes

3. Have the authors made all data underlying the findings in their manuscript fully available?

Reviewer #1: No

Reviewer #2: No

4. Is the manuscript presented in an intelligible fashion and written in standard English?

Reviewer #1: Yes

Reviewer #2: Yes

5. Review Comments to the Author

Reviewer #1: as regard the manuscript " Assessing ascertainment bias in atrial fibrillation across US minority groups type of AF" paroxysmal, persistent or permanent is not included.

has bled score and usage of anticoagulants are not mentioned in data

Reviewer #2: The manuscript PONE-D-23-40160 presents a non-interventional cohort study focused on estimating the prevalence and incidence rates of atrial fibrillation (AF) across minority groups in the United States (US), to aid in diversity enrollment for randomized controlled trials. To achieve this goal, the authors utilized a comprehensive US-based claims database CDM source, encompassing both Asian and Hispanic subgroups.

The study employed both clinical-based and monitoring-based settings within the same population. The authors specifically assessed the impact of ascertainment bias on prevalence rates by calculating magnitude and relative differences across racial and ethnic groups. Notably, the findings revealed significant disparities, particularly among Asian and Hispanic subjects in the overall and stroke cohorts. These differences may be attributed to potential under-ascertainment of AF in the general population.

Considering the study's design, robust statistical analysis, well-structured manuscript writing, effective data presentation, insightful discussion with limitations of this study, and compelling conclusion, I highly recommend this manuscript for publication. The presented findings can contribute to improving the design of clinical trials with real-world data and addressing biases related to health disparities, particularly among underrepresented groups.

6. PLOS authors have the option to publish the peer review history of their article (what does this mean?). If published, this will include your full peer review and any attached files.

Reviewer #1: No

Reviewer #2: **Yes: **Neeraj Sharma

---

## [Author Response · Author response to Decision Letter 0]

20 Mar 2024

Dear Editor & Reviewers,

I hope this letter finds you well. I appreciate the time and effort invested by the editorial team and reviewers in evaluating my Research Article titled "Assessing ascertainment bias in atrial fibrillation across US minority groups" for the PLOS One Journal. I am grateful for the feedback provided, and I would like to submit a rebuttal addressing the comments raised during the review process.

Editor Comments

While the study focused on investigating the presence of ascertainment bias, the study's design lends itself to ascertainment bias; in that those patients with a recent stroke would have increased focus on recognizing AF. I encourage the authors to discuss this oddity and include language or evidence to support the stance that claims data represents real-world data. 

Yes, it is possible that patients with AF given a recent ischemic stroke (AF | stroke & claims) may be different from overall AF population (AF | claims). In the paper we refer to the [AF | claims] group as 'clinical-based detection setting' and the [AF | stroke, claims] group as 'monitoring-based detection setting'. However, since the stroke subgroup is a subset of the overall AF population observed in the claims data (i.e. all the AF stroke patients also met the general inclusion criteria for AF) therefore they should come from the same distribution. In clinical practice, all patients that have an ischemic stroke event receive AF-monitoring/screening, therefore we control for bias in screening in this population. Further evidence of this bias control is the alignment of our results in claims data with previous cited observational studies, where the bias control is achieved through monitoring-based diagnosis methods. We updated the paragraph in the limitations section:

“This study examines ascertainment bias in AF, and the study design controls for such bias, by contrasting the overall population with a cohort of patients with a recent ischemic stroke, for which AF monitoring is part of stroke guidelines. The study relies on claims data to characterize differences, as opposed to observational studies that rely on active monitoring-based diagnosis methods for AF to assess the impact of ascertainment bias in AF differences across racial and ethnic groups.”

Line 111: Ref 27 is repeated. Please edit.

We removed the duplicate reference.

Again, in Line 111, I suggest using “blacks” instead of “African Americans” as “blacks” is more encompassing and would be in keeping with the style of the CMD. Alternatively, provide a working definition of your preferred terms.

We aligned the manuscript to the OMB standards (https://orwh.od.nih.gov/toolkit/other-relevant-federal-policies/OMB-standards), and modified instances of ‘Black’ or ‘African American’ to ‘Black or African American’ and ‘Hispanic’ to ‘Hispanic or Latino’. To align to recent research standards, we removed racial and ethnic terms in noun form. (https://jamanetwork.com/journals/jama/fullarticle/2783090). We updated the manuscript to reflect both OMB standards and adjectival forms.

Line 153: “The patient’s name and geography are...” Are the authors describing the data vendor’s treatment of the claims data or their manipulation of it? Please clarify as the latter situation may be misconstrued as a HIPAA breach. The statement on Line 156 claiming de-identified data must be reconciled with the previous quote.

We adapted the statement to clarify that the data vendor used patient name & geography, not the authors:

“The patient’s name and geography are used by the data vendor to map the patient to one of five race and ethnicity categories…”.

Line 170: Please note that including ICD-10 codes such as I64.x (which could include hemorrhagic strokes) will reduce the specificity of your inclusion. One would not routinely monitor for AF in hemorrhagic strokes. Please clarify or include as a limitation in the discussion. 

We reviewed the ischemic stroke diagnosis codes based on published literature on coding sensitivity and specificity for stroke and stroke risk factors. We acknowledge the potential for miscoding and subsequent risk for misclassification bias in the limitations section:

“There is the potential for miscoding in the atrial fibrillation and ischemic stroke events and subsequent risk of misclassification bias in patient records due to provider coding patterns (e.g., using diagnosis codes to indicate rule-out criterion) or incorrect coding (e.g., data entry errors), which may lead to misclassification of diagnoses or patient characteristics. The algorithms identified for the current study have been previously validated with a focus on precision [42, 44].”

The expression shown on Line 226 is the prevalence ratio (by definition), but it was called “magnitude of difference”, which would invoke the absolute difference. Since this expression is used throughout the paper, it constitutes a major flaw. 

We revised the manuscript and opted for ‘prevalence ratio’ as opposed to ‘magnitude of difference’. Prevalence ratio is a more widely recognized term for describing the formula. The formula, the results, and the conclusion of the manuscript remain the same. 

Reviewer 1 Comments

as regard the manuscript "Assessing ascertainment bias in atrial fibrillation across US minority groups type of AF" paroxysmal, persistent or permanent is not included.

The ICD-10 diagnosis codes that were used for atrial fibrillation (I48*) contain paroxysmal (I48.0) and persistent or permanent (I48.1) atrial fibrillation.

has bled score and usage of anticoagulants are not mentioned in data 

The HAS-BLED score and usage of anticoagulants was not included in the manuscript, in favor of the CHA2DS2-VASc score. The CHA2DS2-VASc score measures Atrial Fibrillation stroke risk, which more closely aligns with the study design. The HAS-BLED score and usage of anticoagulants on the other hand focusses on correctable risk factors for bleeding. Note that both risk scores have a strong overlap in terms of criteria (hypertension, age, stroke & vascular disease).

I have reviewed each comment, and I believe the revisions strengthened the manuscript. 

I would like to express my gratitude for the opportunity to submit this revised manuscript for reconsideration. I believe the revisions adequately address the concerns raised, and I am confident that the manuscript is now in line with the standards and objectives of PLOS One.

Thank you for considering our work for inclusion in your journal. We look forward to your feedback.

Sincerely,

Lars Hulstaert

Johnson & Johnson

---

## [Editor Report · Decision Letter 1]

27 Mar 2024

Assessing ascertainment bias in atrial fibrillation across US minority groups.

PONE-D-23-40160R1

Dear Dr. Hulstaert,

We’re pleased to inform you that your manuscript has been judged scientifically suitable for publication and will be formally accepted for publication once it meets all outstanding technical requirements.

Kind regards,

Daniel Antwi-Amoabeng, MD, MSc

Academic Editor

PLOS ONE

Additional Editor Comments (optional):

Thank you for addressing the suggested revision. The revised manuscript, which is now acceptable for publication.
---

## [Editor Report · Acceptance letter]

3 Apr 2024

PONE-D-23-40160R1 

PLOS ONE

Dear Dr. Hulstaert, 

I'm pleased to inform you that your manuscript has been deemed suitable for publication in PLOS ONE. Congratulations! Your manuscript is now being handed over to our production team.

Kind regards, 

on behalf of

Dr. Daniel Antwi-Amoabeng 

Academic Editor

PLOS ONE